# Adoption of Health Mobile Apps during the COVID-19 Lockdown: A Health Belief Model Approach

**DOI:** 10.3390/ijerph19074179

**Published:** 2022-03-31

**Authors:** Nouf Sahal Alharbi, Amany Shlyan AlGhanmi, Mochammad Fahlevi

**Affiliations:** 1Department of Health Administration, College of Business Administration, King Saud University, Riyadh 11451, Saudi Arabia; noufsahal@ksu.edu.sa; 2Department of Health Informatics, College of Health Sciences, Saudi Electronic University, Jeddah 93499, Saudi Arabia; a.alghanmi@seu.edu.sa; 3Management Department, BINUS Online Learning, Bina Nusantara University, Jakarta 11480, Indonesia

**Keywords:** adoption, MHealth app, mobile applications, COVID-19, coronavirus, Saudi Arabia, Health Belief Model

## Abstract

This study aimed to investigate the adoption of the Sehha, Mawid, and Tetamman mobile health applications during the COVID-19 pandemic in Saudi Arabia. The present study investigated factors influencing app use intention based on the Health Belief Model (HBM) approach. This study was conducted using a sample of 176 participants from the Riyadh and Makkah regions during the lockdown in May 2020. This study uses structural equation modeling for data collected using SmartPLS 3.3.9 (GmbH, Oststeinbek, Germany) to examine the effect of constructs on the model. The most important predictor was the perceived benefits of the mobile health apps, followed by self-efficacy. The perceived barriers and cues to action have no significant effect on behavioral intention. The perceived benefits and self-efficacy as keys can provide an overview to the government and to health organizations for taking into account the most important factors of the adoption of mobile health apps, meaning that the developer must adjust to the characteristics of the community of people that need applications that provide many benefits and have an impact.

## 1. Introduction

Coronavirus disease (COVID-19) is an infectious disease caused by the SARS-CoV-2 virus. Most people who contract COVID-19 will experience mild to moderate symptoms and will recover without special treatment. However, some people will experience severe pain and require medical assistance [1]. The virus can be spread from the mouth or nose of an infected person through tiny fluid particles when the person coughs, sneezes, talks, sings, or breathes. These particles can range from larger droplets from the respiratory tract to smaller aerosols. A person can get infected when they breathe air that contains the virus if they are in close proximity to someone who is already infected with COVID-19. A person can also be infected by touching their eyes, nose, or mouth after touching a contaminated surface. Viruses are easier to spread indoors and in crowded places [2].

A number of policies were taken by Saudi Arabia to cope with the spread of the coronavirus that has spread in the country [3]. As a country that has been visited by many foreign nationals from various parts of the world to perform Hajj and Umrah, Saudi Arabia has made a number of policies related to anticipating the spread of the COVID-19 virus [4]. Saudi Arabia suspended all international flights for two weeks in 2020. The policy was carried out following the increasingly widespread outbreak of the coronavirus in the world. The suspension took effect on 15 March 2020, and earlier that week all of Europe, 12 other countries in Asia, including Indonesia, and Africa were included in the list of countries prohibited from visiting Saudi Arabia [5].

Saudi Arabia’s efforts to control the spread of COVID-19 are gaining global attention. This comes after researchers across the Kingdom highlighted the success of a series of measures adopted in recent months by Saudi Arabia. Saudi Arabia is ranked second—on par with Hungary—on the latest Nikkei COVID-19 Recovery Index [6]. The report was released in August 2021. That is, of 120 countries, based on their performance in terms of infection management, vaccine rollout, and social mobility, Saudi Arabia received a score of 72—only one point behind China’s top ranking. The acceleration of vaccine delivery through more than 585 vaccination centers spread throughout the country greatly contributed to this classification. Individuals have the freedom to decide when and where to receive vaccines. Since the COVID-19 pandemic began in early March 2020, Saudi Arabia has rolled out various health protocols that have reduced the number of infections from around 5000 in mid-June to just hundreds today. The World Health Organization (WHO) praised the Kingdom’s approach. Meanwhile, many heads of state referred to the Saudi example as a “success story” during the G20 leaders’ summit on 22 November 2020 [7].

Saudi Arabia’s success in dealing with the spread of the COVID-19 virus cannot be separated from the role of several health mobile apps owned by the Saudi Arabian government and health organizations in Saudi Arabia [8]. Governments and health organizations began using digital tools to prevent, manage, and control the spread of COVID-19. These technologies include online dashboards, web portals, robots, and mobile health applications (mHealth apps) [9]. Several apps have been created to limit the risk of COVID-19 spreading after the lockdown restrictions are lifted. Contact tracing is at the heart of the app. The potential routes of viral transmission in a community can be investigated using contact tracing to isolate and assist individuals who may have come into touch with someone infected with COVID-19 [10]. Citizens can help limit the spread of COVID-19 by using applications that track contacts with affected individuals and provide advice on how to avoid infection [11].

In 2019, it was predicted that Saudi Arabia had over 40 million mobile cellular users, with 95 percent of them having Internet access [12]. The Saudi Ministry of Health (MOH) created three mHealth apps for public health reasons in response to the increased usage of smartphones by the general public [13]. The first mHealth app, “Sehha” was released in December 2017 and was essential in the early stages of the pandemic. This was because it was created to give long-term and innovative solutions that allowed customers (app users) to obtain healthcare and preventative care at home via digital and telecommunication services, including virtual clinics and teleconsultation. This level of accessibility was critical, especially for monitoring patients’ medical conditions without placing them at danger of infection as a result of being physically present at a clinic.

The second service was a self-assessment function on “Mawid”, the Ministry’s official appointment booking app, which was launched in 2018. This app was created to offer app users with a list of self-assessment questions to identify if they have developed any indications or symptoms related to the infection. In April 2020, the third mHealth app “Tetamman” was released. It was designed particularly to determine whether the users were either unconfirmed infected and required social distance or confirmed infected individuals who required 14 days of quarantine. As a result, they were given a timer and other protective and helpful measures. There was also an optional feature that allowed the contact numbers to reach the authorities considerably faster in the event of an emergency. The presence of a mobile health app is not only a form of technological development, but also for the convenience of getting health services. Through this digital health application, doctors can diagnose based on the results of patient consultations. In addition, there are several other reasons why mobile health apps are an important application for the community to have [14].

Mobile health apps are one of the keys to making it easier for people to get health services online [15]. A consultation with a doctor of choice can be through this application without having to come to the clinic or hospital. The operational system is made easy and simple so that it does not make it difficult for users to access the features in it. Comprehensive health services can be obtained from consultation and treatment to pre- and post-treatment examinations [16]. Reducing mobility during the pandemic is indeed enough to change the pattern of society, especially in terms of health. In the past, it was easy to come to the clinic or hospital, nowadays there are limits to visits per day [17].

Mobile health apps can be useful healthcare aids, especially in underdeveloped countries where healthcare systems are underprepared to deal with the rising number of COVID-19 patients [18]. Previous research has found that two of the most important benefits of employing mobile health systems are increased patient safety and cost savings [19,20]. Nonetheless, the advantages of mHealth apps can only be realized if the user accepts and uses them. Given that the disease is new, there is currently a window of opportunity for efficient usage of mHealth apps during the pandemic [21,22]. During the COVID-19 lockdown in Saudi Arabia, this study attempted to analyze the adoption of health mobile apps based on the Health Belief Model. According to the HBM, two cognitive processes determine an individual’s health behavior in response to a threat: how seriously a person views the threat’s effects and how efficient and suitable the protective behavior is [23].

## 2. Materials and Methods

This study was designed using a behavioral-based model to better understand and predict the adoption of mobile health apps by examining the influence of four variables. Our study was conducted in Saudi Arabia, one of the top few countries with a successful handling of COVID-19 cases and a significant reduction rate in the world. At the time of this study, all three mobile health apps were present in Saudi Arabia. An online survey was administered to respondents, aged 18 to 50. We conducted an online survey from 7 to 17 May 2020 due to the social distancing regulations. The study protocol and informed consent form were in accordance with the standards of the Research Ethics Committee, Deanship of Scientific Research at Saudi Electronic University and received approval (SEUREC-CHS20107). This research has a sample size of 176 participants. This study was conducted among the general population of two major provinces in Saudi Arabia: Riyadh, the capital city in the eastern region, and the Makkah region.

In this study, the measurement of HBM construction followed the recommendations of Walrave, Waterloos, and Ponnet [23] who built the HBM construction based on Champion’s [24]. This study has differences with previous research conducted by Walrave et al. in Belgium because this study did not include respondent characteristics, like age as a construct, which affects the perceived susceptibility and perceived severity because in previous studies it was known that the two constructs had no relationship with behavior intention in using mobile health apps.

This study uses a 5-point Likert scale ranging from strongly disagree to strongly agree. This study uses structural equation modeling for data collected using SmartPLS (GmbH, Oststeinbek, Germany) to examine the effect of constructs on the model [25]. First, we measure the level of validity on each item and whether it can describe the latent variable by evaluating the outer loading and AVE, after that we check the reliability value of each construct using Cronbach’s Alpha, and then we check the discriminant validity using the Fornell–Larcker Criterion—this procedure is included in the PLS-Algorithm [26]. In the final stage of data analysis, we use bootstrapping to determine the effect between the constructs on this HBM model. In the complete PLS SEM bootstrapping method, all values that can be analyzed in the partial least squares analysis are bootstrapped to produce the probability values [27]. We used a process for assessing the level of significance or probability of direct effects, indirect effects, and total effects. In addition, bootstrapping the non-parametric procedure can also assess the level of significance of other values, including the R square and adjusted R square, F square, outer loading, and outer weight. The statistical significance level was assumed for all estimations as a *p* ≤ 0.05 [28].

A total of 176 participants completed the survey, most of whom were Saudi citizens (94.3%), female (68.1%), aged between 18 and 30 years (37.3%), and were bachelor’s degree holders (57.6%). The majority of the study sample (89%) were aware of all three apps. However, only a little over 49% of the participants had downloaded at least one of the health apps (58%). This figure reflected the low adoption level of the app services, such as video calls and COVID-19 symptom checkers, which ranged from 6% to 20%. Compared to the number of app downloaders to users, we found that the user rates for Tetamman, Sehha, and Mawid were 45%, 88%, and 92%.

## 3. Results

### 3.1. Measurement of Model Assessment

A model measurement is needed to determine the relationship between the latent variables and each indicator. At this stage, it is necessary to know the value of the average variance extracted (AVE), Outer Loading, Cronbach’s Alpha, and Composite Reliability. Figure 1 shows the measurement model in this study. 

The outer loading of each item and the AVE on each variable are used to measure the validity of the latent variable with the item indicators. The outer loading measured the correlation between the items and latent variables in the most common theory as >0.7, while the AVE measured the average variance on each variable. The latent variable is generally said to be valid when the AVE value is >0.5 [26]. In the next stage, a reliability test is done by looking at Cronbach’s alpha and the composite reliability to determine the reliability of each variable. In general, if both of them should have a value > 0.7, then it is stated that a variable is reliable [29]. In Table 1 the value of the AVE, Outer Loading, Cronbach’s Alpha, and Composite Reliability will be presented.

In Table 1, it is known that all the variables have met the AVE criteria, but not all items meet the outer loading standard criteria. The variable perceived benefits item PBE1 has a value of 0.552 < 0.7, so it can be said that this item does not pass the validity test and will be deleted in the next stage because it does not meet the standard criteria. Based on the table above, it is known that all the variables have met the reliability requirements because they have Cronbach’s alpha values and a composite reliability > 0.7, so it can be said that all the variables in this research model have passed the reliability test [28].

The discriminant validity is the stage at which the measurement results of a concept are able to distinguish themselves from the results of other concepts. Theoretically, it must be different. Discriminant validity is also part of the outer model. The condition for fulfilling this discriminant validity condition is that the results in the combined loading and cross-loadings view show that the loading to another construct’s cross-loading is a lower value than the loading to a variable construct [26]. In the results of Table 2, it is known that all the variables have no symptoms of cross-loading between the variables.

### 3.2. Path Analysis

This stage is the final analysis stage, which uses bootstrapping procedures for path analysis to determine the effect of the variables in this research model. The calculations were performed using a bootstrap of 5000 samples and 5000 permutations with a significant *p*-Value of 5%. A *p*-Value < 0.05 confirms that there is a significant effect between the two variables.

In Figure 2 it is known that the PBE variable has the highest *t*-value compared to other variables. This shows that the PBE variable has the greatest level of influence on BI in this model. The SE variable is the variable that has the second largest *t*-value in this model against BI, while the remaining two variables, namely PBA and CTA, have a fairly low *t*-value for BI.

Table 3 explains that from the model initiated in this study, there are only two accepted hypotheses, namely hypothesis 1 and hypothesis 4. These results explain that the perceived benefits and self-efficacy variables are the strongest factors that influence the behavioral intention of the Saudi community in adopting the mobile health apps during the lockdown in Saudi Arabia.

## 4. Discussion

To the best of our knowledge, this is the first study to provide a comprehensive picture of the adoption of mHealth apps during the COVID-19 lockdown in Saudi Arabia. The results of this study are consistent with previous research conducted by Walrave [23] in Belgium, which found the same results—i.e., that the perceived benefits and self-efficacy were the biggest variables that influenced the behavioral intention to adopt the mobile health apps. These results explain that there are similarities in the characteristics of the community in adopting these mHealth apps. The presence of health applications presents sophisticated breakthroughs that can revolutionize the health industry today [30]. The existence of this application can cut the barriers of distance and time, which can also mean that many lives can be saved. Not to forget, health applications also have a big influence on improving healthy lifestyles that develop in society. In addition, easy access to health information, apart from providing education, also increases public awareness of healthy living. Equally important, mobile applications for health that are widely circulated in Saudi Arabia can also act as assistants to help sustain patient care. Tracing during a pandemic is an important lesson in how health applications are very useful for the government to deal with extraordinary events such as the COVID-19 pandemic. However, people also need to be careful. Even if it helps society a lot, sharing data with health apps should be met with caution. Make sure the application used is valid or developed by a trusted company, because medical records are sensitive and confidential information [31].

The role and efficacy of the application in managing COVID-19 should be defined, since the perceived usefulness is the most essential component when it comes to the intended use of the program. The need for tracing contacts and reporting suspected viral exposure must therefore be discussed and displayed when deploying a COVID-19 tracking application. Several talks have been given to explain respiratory viral aerosolization and how the COVID-19 application may be used to map close individual contacts with a high risk of infection. Additional options can be incorporated to inform and help users in order to preserve perceived benefits over time [32].

The government of Saudi Arabia optimized public health protection through the development of mobile health apps. The government also asked for the participation of citizens to download and use the technology. This is because this technology is expected to simplify and shorten the flow of information, be effective, and be adaptively adopted in various sectors. In addition, digital technology plays an important role in guiding distances or locations, which is an important feature in controlling the spread of COVID-19. These three applications are no exception. The application of digital technology to support the implementation of health protocols and 3T (Testing, Tracing, Treatment) is one of the crucial efforts in controlling the pandemic. With that, people need to start getting used to digital technology in their daily lives, one of which is the use of mobile health apps [33].

The perceived benefits and self-efficacy as keys can provide an overview to the government and to health organizations in taking into account the most important factors of the adoption of mobile health apps [23,24], meaning that the developer must adjust to the characteristics of the community of people that need the applications that provide many benefits and have an impact on them. The impact should be directly on the individual, given that public awareness is a very important self-efficacy factor that needs to be built-in by all stakeholders—and is not only the task of the government. The problem of pandemics and their handling is a shared responsibility, and the successful adoption of mobile health apps during a pandemic is expected to assist the government in mapping and providing medical assistance to the right people.

## 5. Conclusions

Governments are considering contact tracing apps as an important part of their lockdown exit tactics during the COVID-19 lockdown. At this time, it is important for a country to develop mobile health apps, and digital health service developers should begin to formulate tools that can transmit patient medical data and give alert signals to designated hospitals or relatives. The goal is that needed assistance can be given immediately, especially during a pandemic. Health services in remote areas can also be assisted with telemedicine. The perceived benefits and self-efficacy are the strongest factors in the adoption of mobile health apps in Saudi Arabia, given that the government and health organizations need to pay attention to these two variables in developing applications. However, this study has several limitations. First, the data were cross-sectional; therefore, biases can result from non-responses and self-reported data. Second, since the number of actual users of the Tetamman app was considerably lower compared to the other two, the generalizability of the findings is limited. Therefore, future research should examine the efficacy of these apps and their potential for improvement.

## Figures and Tables

**Figure 1 ijerph-19-04179-f001:**
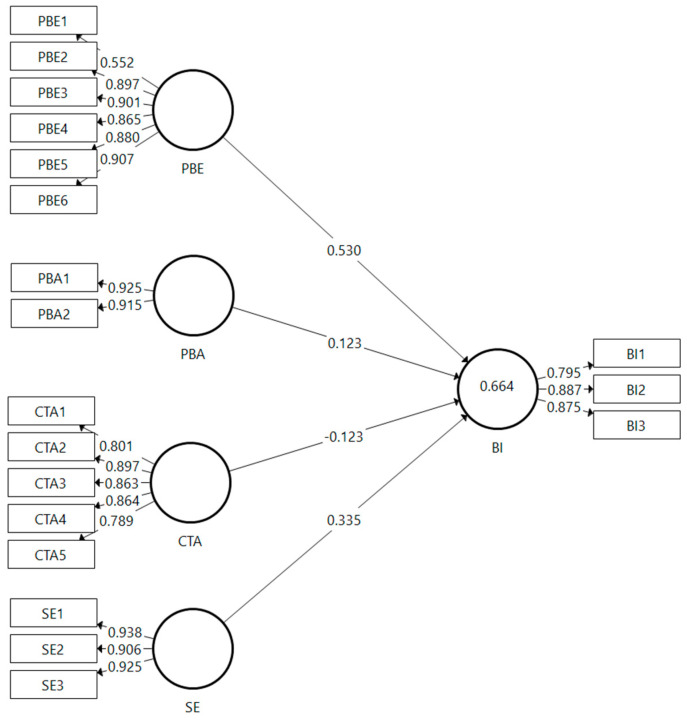
Model Assessment.

**Figure 2 ijerph-19-04179-f002:**
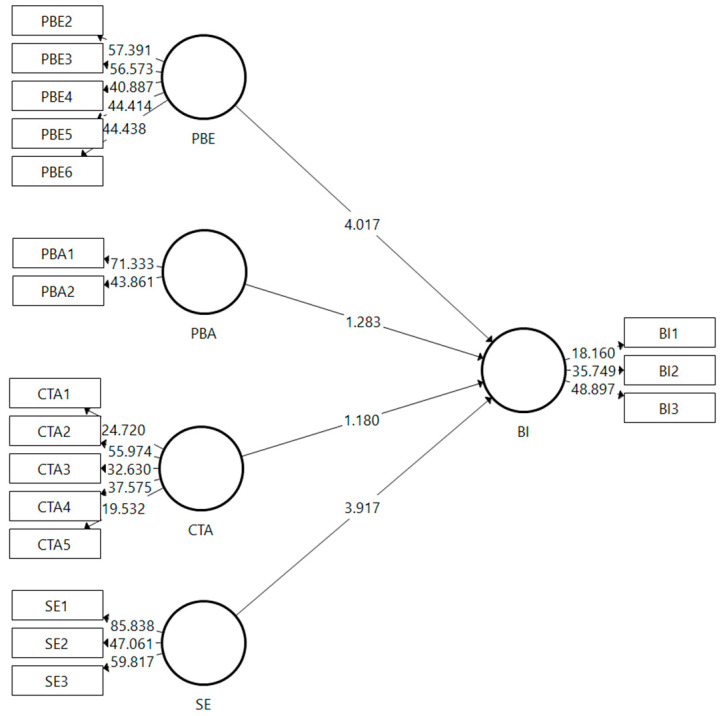
Bootstrapping.

**Table 1 ijerph-19-04179-t001:** PLS-Algorithm.

Variable & Item	AVE	Outer Loading	Cronbach’s Alpha	Composite Reliability
Behavioral Intention	0.728		0.813	0.889
BI1		0.795		
BI2		0.887		
BI3		0.875		
Perceived Benefits	0.711		0.913	0.935
PBE1		0.552		
PBE2		0.897		
PBE3		0.901		
PBE4		0.865		
PBE5		0.880		
PBE6		0.907		
Perceived Barriers	0.847		0.819	0.917
PBA1		0.925		
PBA2		0.915		
Cues to Action	0.712		0.898	0.925
CTA1		0.801		
CTA2		0.897		
CTA3		0.863		
CTA4		0.864		
CTA5		0.789		
Self-Efficacy	0.852		0.913	0.945
SE1		0.938		
SE2		0.906		
SE3		0.925		

**Table 2 ijerph-19-04179-t002:** Discriminant Validity (Fornell–Larcker Criterion).

Constructs	BI	CTA	PBA	PBE	SE
BI	0.853				
CTA	0.687	0.844			
PBA	0.719	0.809	0.920		
PBE	0.786	0.824	0.841	0.843	
SE	0.729	0.820	0.748	0.761	0.923

**Table 3 ijerph-19-04179-t003:** Path Coefficients Estimation.

Hypothesis	Variable	*t*-Values	*p*-Values	Supported
H1	PBE → BI	4.017	0.000	Yes
H2	PBA → BI	1.283	0.200	No
H3	CTA → BI	1.180	0.239	No
H4	SE → BI	3.917	0.000	Yes

## Data Availability

Not applicable.

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
