# Peer review of "Adoption of Health Mobile Apps during the COVID-19 Lockdown: A Health Belief Model Approach"

_ijerph, 2022, doi:10.3390/ijerph19074179_

Round 1
Reviewer 1 Report
Thank you for the opportunity to review this manuscript. This is an interesting study on an important topic such as the use of the mobile applications, which is rising. First of all, I believe this is not a case report, rather observational study. therefore the type of article should be changed accordingly and the study design should be added to the title. Other comments:
Explain in the abstract what are Sehha, Mawid and Tetamman
Improve conclusion in the abstract so it could be more generalized for all countries
explain in the methods the income in your value compared to euro or dollar
Author Response
I thank you for your valuable input to make this research even better.
We have corrected our research based on the comments and inputs you have given as a whole starting from the abstract to the conclusion, we changed it quite majorly according to the direction you gave
Reviewer 2 Report
The paper aims to provide an interesting analysis of the public awareness, uses, and acceptability of the government’s m Health apps during the COVID-19 pandemic in Saudi Arabia.
The study was conducted on a cross-sectional online survey based on convenience sample of 554 participants from 15 the Riyadh and Makkah regions during the lockdown in May 2020.
The strength of this study is an exciting analysis helpful approach to improve social awareness, trust, and more information on the use of a public’s app
Although previous works on the same topic are mentioned, it would be interesting to link them better in the paper.
The paper shows a controversial point in 48 because the word launched appears unclear. After all, the app was changed or updated, not established concerning COVID 19.
The demonstrations are carried out in a logical way.
The article's title is well chosen, short enough yet clear enough and it highlights the purpose of the study and its potential applications.
The abstract summarizes well the main points treated in the article.
Conclusions are correct and justified by the demonstration
The article has the potential to be published, but authors have to operate some revisions to give more consistency to the manuscript.
Table 1 has to be better presented in clearer way.
From 99 to 105 could also be presented with a Table.
Table 2 11 and 19 the signs seems not correct.
170-171 the concept is unclear and well linked to the rest.
Author Response
I thank you for your valuable input to make this research even better.
We have corrected our research based on the comments and inputs you have given as a whole starting from the abstract to the conclusion, we changed it quite majorly according to the direction you gave
We have changed the tables and figures in this study according to your direction
Reviewer 3 Report
This paper discuss the public acceptance towards government health mobile apps during the COVID-19 lockdown. For this reviewer, the contribution of this research is not clear. The manuscript needs considerable improvement, mainly in its organization. There are several grammar, punctuation, and speeling errors. Besides, typo errors.
Here are some suggestions for improvement:
1) The introduction is brief and superficial. Tables/figures should be in other sections.
2) The materials and methods section leaves a lot to be desired. How was the data handled? Who are the subjects of research? There are some issues to be addressed.
3) The results are not encouraging and do not seem to be in line with reality. A comparison with other current state of the art studies is necessary.
Author Response

(The authors gave the same response as above.)

Reviewer 4 Report
This paper presents a study that aimed to assess three mHealth apps during the COVID-19 lockdown in Saudi Arabia.
Suggestions and questions (answers can/should be used to improve the paper):
1. The reported results (e.g., 4.21±0.62) in the abstract do not make sense if the range values are not known.
2. Screenshots of the apps should be outside the table 1 as a figure.
3. Column headers should be added in Table 1.
4. How did the researchers define the questionnaire? Why not use questionnaires already well-defined in the literature? For example, UEQ - https://www.ueq-online.org/ , System Usability Scale (SUS), etc. This should be explained.
5. The third part of the questionnaire should be presented. Suggestion: present it as a supplementary material.
6. How many participants answered the third part of the questionnaire? 475?
7. Limitations should be acknowledged in the discussion section. Conclusion section should present future work.
8. Explain/detail/unpack this sentence "future research should examine the efficacy of these apps and their potential for improvement"
9. What can be learned from the results of this study? What [new] directions can be suggested to the app developers? Discuss this.
Specific comments:
- mobile cellular -> smartphones?
- line 126: This figure reflected... - what figure?
Author Response
I thank you for your valuable input to make this research even better.
We have corrected our research based on the comments and inputs you have given as a whole starting from the abstract to the conclusion, we changed it quite majorly according to the direction you gave
We have changed the tables and figures in this study according to your direction
we continue to use the phrase mobile cellular according to the source we took from worldbank